# Relationship between plasma uric acid levels, antioxidant capacity, and oxidative damage markers in overweight and obese adults: A cross-sectional study

Natalia Albuja-Quintana[1], Aida M. Chisaguano-Tonato[1], Maria E. Herrera-Fontana[1], Sandra Figueroa-Samaniego[2], José M. Alvarez-Suarez[3,4]*

1 Colegio de Ciencias de la Salud, Nutrición y Dietética, Universidad San Francisco de Quito USFQ, Quito, Ecuador, 2 Unidad Académica de Ciencias Químicas y de la Salud, Universidad Técnica de Machala, Machala, Ecuador, 3 Departamento de Ingeniería en Alimentos, Laboratorio de Investigación en Ingeniería en Alimentos (LabInAli), Colegio de Ciencias e Ingenierías, Universidad San Francisco de Quito USFQ, Quito, Ecuador, 4 Laboratorio de Bioexploración, Colegio de Ciencias Biológicas y Ambientales, Universidad San Francisco de Quito USFQ, Quito, Ecuador

* jalvarez@usfq.edu.ec

**Data Availability Statement:** The underlying minimal data for this study can not be shared

## Abstract

Obesity, a complex metabolic chronic illness, is commonly accompanied by an excessive production of free radicals, which influences the development of its comorbidities. Uric acid is commonly linked to pro-oxidant effects on human health. Though recent evidence suggests its potential antioxidant properties, it is possible that the increase in circulating uric acid levels is an adaptive protective response against the detrimental effects of excess free radicals and oxidative stress present in obese individuals. Hence, the purpose of this study was to evaluate the association between antioxidant capacity and oxidative damage markers with uric acid levels in overweight and obese individuals who live at sea level. This cross-sectional study included 93 adult volunteers (28 men and 65 women) from the city of Machala, El Oro, Ecuador, divided into three study groups according to Body Mass Index (normal weight, overweight, and obese). Sociodemographic characteristics, lifestyle elements, and physical measurements were evaluated, and blood samples were collected from all participants. Antioxidant and oxidant markers, including Radical-Scavenging Activity Assay (DPPH), Ferric Reducing Ability of Plasma (FRAP), Catalase (CAT) activity, Thiobarbituric Acid Reactive Substances (TBARS), and Protein Thiol Groups (SH groups), were assayed in the plasma samples. Coefficients of correlations and linear regression models were applied to evaluate the association between antioxidant/oxidant parameters and plasma uric acid levels. The antioxidant capacity, as measured by FRAP scavenging and CAT, was significantly higher in the obese group compared to the normal weight group, and uric acid levels presented a significant positive association with FRAP (b: 0.578, R: 0.459, p: 0.003) and CAT activity (b: 1.326; R: 0.432, p: 0.005) in overweight and obese individuals. Consequently, the presented evidence supports the potential antioxidant role of uric acid in the pathogenesis of obesity, contributing to our understanding of oxidative stress and inflammation that characterizes this illness.

publicly due to privacy concerns regarding the confidentiality of participants. Data are available from Comité de Ética de Investigación en Seres Humanos (CEISH) (https://www.usfq.edu.ec/es/comite-de-etica-de-investigacion-en-seres-humanos-ceish) via email (ceishusfq@usfq.edu.ec) or telephone ((+593) 02 506 1700 ext. 1149) for researchers who meet the criteria for access to confidential data. Access to the data may be granted upon request to the Principal Investigator, subject to prior authorization from from Comité de Ética de Investigación en Seres Humanos (CEISH). Requests may also be addressed to Professor José M. Álvarez Suárez via email (jalvarez@usfq.edu.ec) and must include details about the intended use of the data. All requests will be reviewed and evaluated by CEISH before approval is granted.

**Funding:** This work was supported by Universidad San Francisco de Quito (Grant Number: 15748) and by Grants from the School of Public Health of the Universidad San Francisco de Quito, Quito, Ecuador. The funders had no role in study design, data collection and analysis, decision to publish, or preparation of the manuscript.

**Competing interests:** The authors have declared that no competing interests exist.

## Introduction

Obesity is a major public health problem and is a complex metabolic chronic illness determined by an elaborate interaction between genetic and environmental factors. As a result, it is commonly accompanied by an excessive production of free radicals (reactive oxygen species, shortened to ROS), which in turn can damage proteins, lipids, and nucleic acids. The balance in this oxidant/antioxidant activity influences the pathogenesis of obesity [1] and its metabolic comorbidities, like insulin resistance, type 2 diabetes, dyslipidemia, hypertension, cardiovascular disease, and fatty liver disease [2,3].

Uric acid is the final product derived from purine metabolism. Uric acid has primarily been investigated for its pro-oxidant effects on human health, and it has been linked to comorbidities like nephropathy and metabolic diseases [4]. Nevertheless, current evidence centers around the antioxidant properties of this compound and the paradox of its duality with prooxidant effects [5–10]. In this regard, uric acid is considered an important scavenger of ROS, as it represents around 60% of the plasmatic scavenger capacity [11]. There have been several potential explanations for this phenomenon. For instance, the protective effects of uric acid may result from a gradual increase in its plasmatic levels, whereas its chronic elevation represents a risk factor for illness [9]. Additionally, it is hypothesized that uric acid has antioxidant properties in a hydrophilic medium, such as plasma, and a pro-oxidant effect inside the cells [6]. Therefore, it is possible that the increase in circulating uric acid levels is an adaptive response to protect against the detrimental effects of excess free radicals and oxidative stress.

In this way, several studies have demonstrated its importance in promoting high antioxidant capacity and reduced oxidative damage, even when illnesses are already present. Thus, in renal hemodialysis patients, high levels of uric acid are related to less oxidative damage and better nutritional status [12]. In addition to this, the incidence of metabolic syndrome in healthy participants is negatively associated with the percentage change in serum uric acid levels. The inverse correlation between high-sensitivity C-reactive protein (hs-CRP) and uric acid levels explains this association [9]. Likewise, patients in the acute phase of an ischemic stroke and low uric acid levels present unfavorable functional outcomes, attributing to uric acid a neuroprotective effect, defending neurons from free radical-mediated damage and endothelial dysfunction due to oxidative stress [8].

Particularly, and in relation to the scope of this study, it has been suggested that in patients with obesity, elevated uric acid levels act as a protective adaptative response to counteract the excessive free radical production and systemic oxidative damage [11–14]. Hence, studies have shown that in obese subjects, serum uric acid level is positively correlated with plasmatic ferric-reducing antioxidant potential (FRAP) and 2,2-Diphenyl-1-picrylhydrazyl radical activity (DPPH), both of which measure antioxidant capacity [1,11–13]. Therefore, the purpose of this study was to evaluate the association between antioxidant capacity and oxidative damage markers with uric acid levels in overweight and obese young adults residing in Ecuador's coastal region.

## Materials and methods

### Subjects and study design

Ninety-three volunteers (28 men and 65 women, aged 22.24 ± 4.5 years) from the city of Machala in the province of El Oro, Ecuador, were included in this study.

Between January 9th and February 14th, 2020, biological samples, sociodemographic data, lifestyle factors, and physical measurements were collected from each selected participant. The inclusion criteria used to accept participants were: adults aged 18 to 50 years living in the

selected areas of Machala from birth until the time of the study, who agreed to participate and signed the informed consent. Volunteers could not be pregnant or suffer from chronic diseases (diabetes, cancer, or cardiovascular diseases) or immunological diseases (autoimmune or immunosuppressive). Additionally, subjects who had spent time in places with a difference in altitude greater than 2,000 meters were not included.

Participants were divided into three study groups according to Body Mass Index (BMI, kg/m$^2$), specifically normal weight (NW, 18.5 to 24.9 Kg/m$^2$; n = 51), pre-obese or overweight (OW, 25 to 29.9 Kg/m$^2$; n = 27) and obese (O, $\geq$ 30 Kg/m$^2$; n = 15) [15]. The study design was cross-sectional, and it was conducted in accordance with the principles of the Declaration of Helsinki, as revised in 2000. The protocol was approved by the Human Research Ethics Committee (CEISH) of the Universidad San Francisco de Quito (Code P2018-176E), Ecuador, and registered in the General Coordination of Strategic Health Development of the Ministry of Public Health of Ecuador with protocol code MSPCURI000308-2. Written consent was obtained from each participant. To analyze and present the findings of this study, we accessed a dedicated database created specifically for this research between January 2023 and July 2024. To maintain participant privacy and confidentiality, each individual was assigned a unique identifier code throughout the data analysis process.

## Assessment of study participant characteristics

Sociodemographic characteristics captured details including age, gender, place of residence, and level of education. Lifestyle elements brought into the analysis included smoking behaviors, adhering to the NCHS's categorical divisions: never or current smokers. Alcohol consumption delineations encompassed non-drinkers (no consumption each week). Physical activity was classified into five groups (sedentary, light activity, moderate activity, high activity, and vigorous activity). Food consumption was evaluated by a short FFQ with general food groups included [16].

Physical measurements such as weight (kg), height (m), BMI (kg/m2), waist circumference (WC; cm), and hip circumference (HC; cm), were determined. The percentages of body fat, visceral fat, and muscle were determined using a bioimpedance scale (OMRON HBF-514C). During the anthropometric measurements, all participants were barefoot and clothed appropriately. Systolic and diastolic blood pressure (SBP and DBP) was measured in the individuals after setting and resting for 10 minutes using a digital apparatus. The values were expressed in mmHg.

## Sample collection

Blood samples (15 mL) were collected from overnight fasting subjects by antecubital venipuncture into a sodium citrate vacutainer (BD Vacutainer CPTTM). The blood samples were centrifuged for 10 min at 4,000 rpm and 4 ˚C. Plasma was collected and isolated from blood samples taken from volunteers and was stored at -80˚C for biochemical studies. To maintain sample integrity and prevent degradation from repeated freeze-thaw cycles, plasma samples were divided into 0.5 mL aliquots. For each analytical run, the necessary number of aliquots was thawed at 4˚C in the dark to minimize temperature fluctuations and light exposure.

## Plasma biochemical analysis

The enzymatic test kits were used to determine plasma glucose (GOD-PAP), glycosylated hemoglobin (HbA1c), total cholesterol (CHOD-PAD), triglycerides (GPO-PAP), HDL cholesterol (HDL-C), LDL cholesterol (LDL-C), creatinine, and uric acid (UOD-PAP). All commercial kits were purchased from Valtek S.A, Macul, Chile, and the analyses were conducted using Roche/Hitachi automated clinical chemistry analyzers.

## Antioxidant and oxidant markers

**Radical-Scavenging Activity Assay (DPPH).** The DPPH scavenging assay was performed according to the colorimetric method described by Prymont-Przyminska et al. [17] with slight modifications, and results were expressed as inhibition percentages {% = [(CA—SA) / CA) × 100]; CA: Absorbance of DPPH solution with methanol and SA: Absorbance of DPPH solution in the sample solution (plasma)}.

DPPH solution (CA) was prepared by mixing 3.2 mg of DPPH radical with 100 mL of absolute methanol, and this solution was adjusted to a maximum absorbance of 0.9 ±0.02 to 517 nm. Briefly, 20 µL of plasma was added to 780 µL of DPPH solution (SA). The sample was mixed and incubated for 15 min at room temperature in complete darkness. Then, the sample was centrifuged to 10,000 rpm for 10 min. Absorbance readings were taken at 517 nm against the blank. The blank was composed of 800 µL of absolute methanol. All analyses were performed in triplicate.

**Ferric Reducing Ability of Plasma (FRAP).** Plasma FRAP levels were measured according to Benzie et al. [18] with slight modifications. FRAP reagent was prepared by mixing 100 mL of acetate buffer (300 mM, pH 3.6) with 10 mL of TPTZ solution (2,4,6-tripyridyl-S-triazine, 10 mM in 40 mM of HCl), and 10 mL of ferric chloride (20mM). Then, 980 µL of FRAP reagent was mixed with 20 µL of plasma. The absorbance readings were taken at 539 nm against the blank. All analyses were performed in triplicate, and the final results were expressed in µM.

**Catalase (CAT) activity.** CAT activity was determined as previously described by Aebi [19] with slight modifications. 10 µL of plasma was mixed with 990 µL of substrate sodium-potassium phosphate buffer (pH 7) and 500 µL of hydrogen peroxide solution (30%). The absorbance was determined at 240 nm twice, at 10 seconds and 70 seconds. All analyses were performed five times. The final results were expressed as international units (IU).

**Thiobarbituric Acid Reactive Substances (TBARS).** Lipid peroxidation was estimated in plasma using TBARS according to Ohkawa et al. [20] with minor modifications. The TBARS reagent was prepared by mixing 0.37g of thiobarbituric acid with 100 mL of HCl solution (2 M), and a trichloroacetic solution at 1% was prepared. 300 µL of plasma was mixed with 700 µL of thiobarbituric reagent and 200 µL of trichloroacetic solution. The mixture was incubated at 85˚C for 20 min in a water bath. Subsequently, the samples were cooled in ice water for 5 min and were centrifuged at 12,000 rpm for 20 min. The reaction product was measured spectrophotometrically at 532 nm. All analyses were performed in triplicate and the results were expressed in µmol TBARS/L.

**Protein thiol groups (SH groups).** Protein thiol groups were assayed in plasma following the method by Asgary et al. [21]. It consisted of the reduction of 5,5′-dithiobis-(2-nitrobenzoic acid) (DTNB, 100 mM) with SDS solution (sodium dodecyl sulfate, 10%), measured at 412 nm. All analyses were performed in triplicate and the results were expressed in µmol P-SH/L.

**Statistical analysis.** Statistical analyses were performed using the Statistical Package for the Social Sciences (SPSS) 22.0 program (IBM Corp.; Armonk, NY, USA). The normality of the distribution was assessed using the Kolmogorov-Smirnov test. Frequencies of categorical values were compared using the $\chi 2$ test. For data that showed compliance with the normal distribution, an analysis of variance (ANOVA) test using the post-hoc Bonferroni or Dunnett test was applied. Results were expressed as n (weighted %) or mean ± SD, respectively.

Spearman or Pearson correlation coefficients were calculated to describe crude and adjusted models, considering the effect of potential confounding factors. To assess the strength of these associations, multivariate linear regression models were applied. A p-value of $< 0.05$ was considered statistically significant.

## Results

### Characteristics of the study population

The baseline characteristics of the study population (n = 93) are described in **Table 1**. The age of the participants ranges between 18 and 43 years, with a sex distribution of 69.4% women and 30.6% men. 64.3% of the participants have higher education. Alcohol consumption occurs in more than 50% of the participants, while tobacco use occurs in 14.7%. The physical activity level that predominates in the three groups is sedentarism and light activity, at 58.4%.

As expected, weight, BMI, body fat percentage, visceral fat percentage, hip and waist circumferences, and systolic blood pressure (SBP) are significantly higher in obese participants when compared with normal-weight subjects.

The values of the biochemical parameters measured in the study are shown in **Table 2**. O group members have significantly lower HDL-C levels (p = 0.024) and higher triglycerides, creatinine, and uric acid levels compared to the NW group, though total cholesterol levels, LDL-C levels, glucose, and Hba1c did not differ significantly between the O and NW groups. Furthermore, higher statistical means were obtained in O subjects for the count of platelets, leukocytes, neutrophils, and eosinophils compared to the NW group. Lastly the O group has a significantly lower lymphocyte count than the NW and OW groups.

**S1 Table** shows the consumption frequency of food groups that could have some effect on the modulation of oxidative status. For instance, white fish consumption is higher in the O

**Table 1. Characteristics of the study subjects according to nutritional status.**

| Characteristic | NW (n = 51) | | | OW (n = 27) | | | O (n = 15) | | | p |
|---|---|---|---|---|---|---|---|---|---|---|
| Weight (kg) | 57.95 | ± | 7.94[a] | 72.50 | ± | 8.98[b] | 85.57 | ± | 17.75[c] | **<0.001** |
| Age (years) | 21.92 | ± | 3.60 | 22.37 | ± | 4.16 | 23.60 | ± | 7.59 | 0.252 |
| BMI (kg/m$^2$) | 22.65 | ± | 1.69[a] | 27.08 | ± | 1.37[b] | 32.99 | ± | 3.39[c] | **<0.001** |
| Muscle (%) | 27.91 | ± | 5.53 | 28.94 | ± | 6.04 | 26.06 | ± | 5.05 | 0.259 |
| Visceral Fat (%) | 4.38 | ± | 1.30[a] | 7.08 | ± | 2.17[b] | 9.14 | ± | 3.46[b] | **<0.001** |
| Body Fat (%) | 31.93 | ± | 7.22[a] | 36.52 | ± | 7.37[b] | 43.05 | ± | 7.17[b] | **<0.001** |
| Waist circumference (cm) | 75.74 | ± | 5.94[a] | 86.20 | ± | 7.59[b] | 96.97 | ± | 13.42[c] | **<0.001** |
| Hip circumference (cm) | 95.59 | ± | 6.45[a] | 104.76 | ± | 7.50[b] | 112.70 | ± | 5.91[c] | **<0.001** |
| Systolic Blood Pressure (mmHg) | 108.69 | ± | 8.31[a] | 112.41 | ± | 11.80[a,b] | 118.67 | ± | 11.25[b] | **0.004** |
| Diastolic Blood Pressure (mmHg) | 70.63 | ± | 10.35 | 72.41 | ± | 6.41 | 75.33 | ± | 11.87 | 0.263 |
| *Demographic and Lifestyle Profile* | | | | | | | | | | |
| Sex, n (%) | | | | | | | | | | 0.141 |
| Female | 40 | (78.40) | | 16 | (59.30) | | 9 | (60.00) | | |
| Level of education, n (%) | | | | | | | | | | 0.631 |
| Higher education | 34 | (66.70) | | 19 | (70.40) | | 7 | (47.70) | | |
| Alcohol consumption, n (%) | | | | | | | | | | 0.905 |
| Yes | 37 | (77.10) | | 20 | (74.10) | | 12 | (80.00) | | |
| Tobacco use, n (%) | | | | | | | | | | 0.121 |
| Yes | 4 | (8.30) | | 6 | (22.20) | | 4 | (26.70) | | |
| Physical Activity, n (%) | | | | | | | | | | 0.480 |
| Sedentary | 13 | (26.50) | | 3 | (11.10) | | 1 | (6.70) | | |
| Light Activity | 15 | (30.60) | | 14 | (51.90) | | 8 | (53.30) | | |

NW: Normal weight; OW: Overweight; O: Obese. Natural logarithms were used for non-normal variables. The ANOVA test with Bonferroni post hoc analysis was applied to quantitative variables. The Chi-square test was applied to qualitative variables. The significance level was established at *p< 0.05. Different superscript letters show significant differences between nutritional status groups. Data are presented as mean ± SD or n (weighted %).

**Table 2. Biochemical parameters according to nutritional status.**

| Parameter | NW (n = 51) | | | OW (n = 27) | | | O (n = 15) | | | p |
|---|---|---|---|---|---|---|---|---|---|---|
| Uric Acid (mg/dL) | 6.27 | ± | 1.25[a] | 7.23 | ± | 1.25[b] | 7.76 | ± | 1.84[b] | **0.001** |
| Glucose (mg/dl) | 80.31 | ± | 8.72 | 82.15 | ± | 8.21 | 80.07 | ± | 7.71 | 0.590 |
| Hba1c (%) | 5.00 | ± | 0.40 | 5.02 | ± | 0.40 | 5.09 | ± | 0.42 | 0.736 |
| Total cholesterol (mg/dL) | 176.24 | ± | 41.35 | 180.25 | ± | 45.75 | 203.67 | ± | 52.40 | 0.107 |
| HDL cholesterol (mg/dL) | 74.88 | ± | 15.42[a] | 74.18 | ± | 15.04[a,b] | 63.25 | ± | 19.74[b] | **0.024** |
| LDL cholesterol (mg/dL) | 99.18 | ± | 26.65 | 95.55 | ± | 22.50 | 112.75 | ± | 42.10 | 0.288 |
| Triglycerides (mg/dL) | 117.84 | ± | 47.62[a] | 128.81 | ± | 47.42[a,b] | 178.77 | ± | 93.71[b] | **0.005** |
| Creatinine (mg/dL) | 0.74 | ± | 0.15[a] | 0.85 | ± | 0.19[b] | 0.87 | ± | 0.18[b] | **0.004** |
| RBC count (×10$^6$/ μl) | 4.63 | ± | 0.56 | 4.89 | ± | 0.57 | 4.93 | ± | 0.42 | 0.058 |
| Hemoglobin (g/dL) | 14.05 | ± | 1.73 | 14.83 | ± | 1.71 | 14.91 | ± | 1.29 | 0.066 |
| Hematocrit (%) | 42.27 | ± | 5.19 | 44.41 | ± | 5.17 | 44.80 | ± | 3.88 | 0.086 |
| Platelet count (x10$^3$ /μL) | 246.72 | ± | 58.6[a] | 261.89 | ± | 47.00[a,b] | 290.57 | ± | 68.04[b] | **0.043** |
| MCV (fl) | 90.77 | ± | 0.26 | 90.84 | ± | 0.17 | 90.79 | ± | 0.25 | 0.541 |
| MCH (pg/mL) | 30.18 | ± | 0.13 | 30.22 | ± | 0.07 | 30.19 | ± | 0.10 | 0.382 |
| Leukocytes (x10$^3$ /μL) | 6.04 | ± | 1.25[a] | 6.61 | ± | 1.34[a,b] | 7.41 | ± | 2.08[b] | **0.014** |
| Neutrophils (%) | 59.41 | ± | 7.66[a,b] | 58.67 | ± | 8.70 [a] | 65.29 | ± | 7.58[b] | **0.038** |
| Lymphocytes (%) | 39.28 | ± | 7.80[a] | 40.30 | ± | 8.86[a] | 33.08 | ± | 6.91[b] | **0.010** |
| Eosinophils (%) | 1.35 | ± | 0.84[a,b] | 1.08 | ± | 0.69[a] | 1.60 | ± | 0.99[b] | **0.037** |
| Basophils (%) | 0 | | | 0 | | | 0 | | | |
| Monocytes (%) | 0 | | | 0 | | | 0 | | | |

NW: Normal weight; OW: Overweight; O: Obese. Natural logarithms were used for non-normal variables. The ANOVA test with Bonferroni post hoc analysis was applied to quantitative variables. The significance level was established at *p< 0.05. Different superscript letters show significant differences between nutritional status groups. Data are presented as mean ± SD.

group than the NW group (46.70% vs 32.60% eat white fish at least once a week), while the remaining food groups did not differ significantly between O, OW, and NW groups.

## Plasma antioxidant capacity and oxidative damage according to nutritional status

The total antioxidant capacity, as measured by FRAP scavenging and CAT activity, was higher in the obese group compared to the normal weight group (p = 0.007 and p = 0.038, respectively) as shown in **Fig 1**. However, no significant differences were observed in these parameters between the OW and O groups. Also, there were no significant differences in levels of plasma DPPH, TBARS, and SH-group between the groups.

## Correlations between antioxidant/oxidant parameters with body composition and biochemical profile

Correlations between all antioxidant and oxidant markers with body composition and biochemical profile were analyzed, as shown in **Table 3**. Regarding antioxidant markers, FRAP, CAT, and DPPH were all positively correlated with TC and TG. Meanwhile, oxidative markers had no common results, where TBARS maintained positive correlations with visceral fat and TC, and negative with HDL, while the SH group and TG were negatively correlated.

**S2 Table** shows the specific correlation analysis of uric acid levels with biochemical parameters and body composition, where three models were applied: a crude model (CM), model I

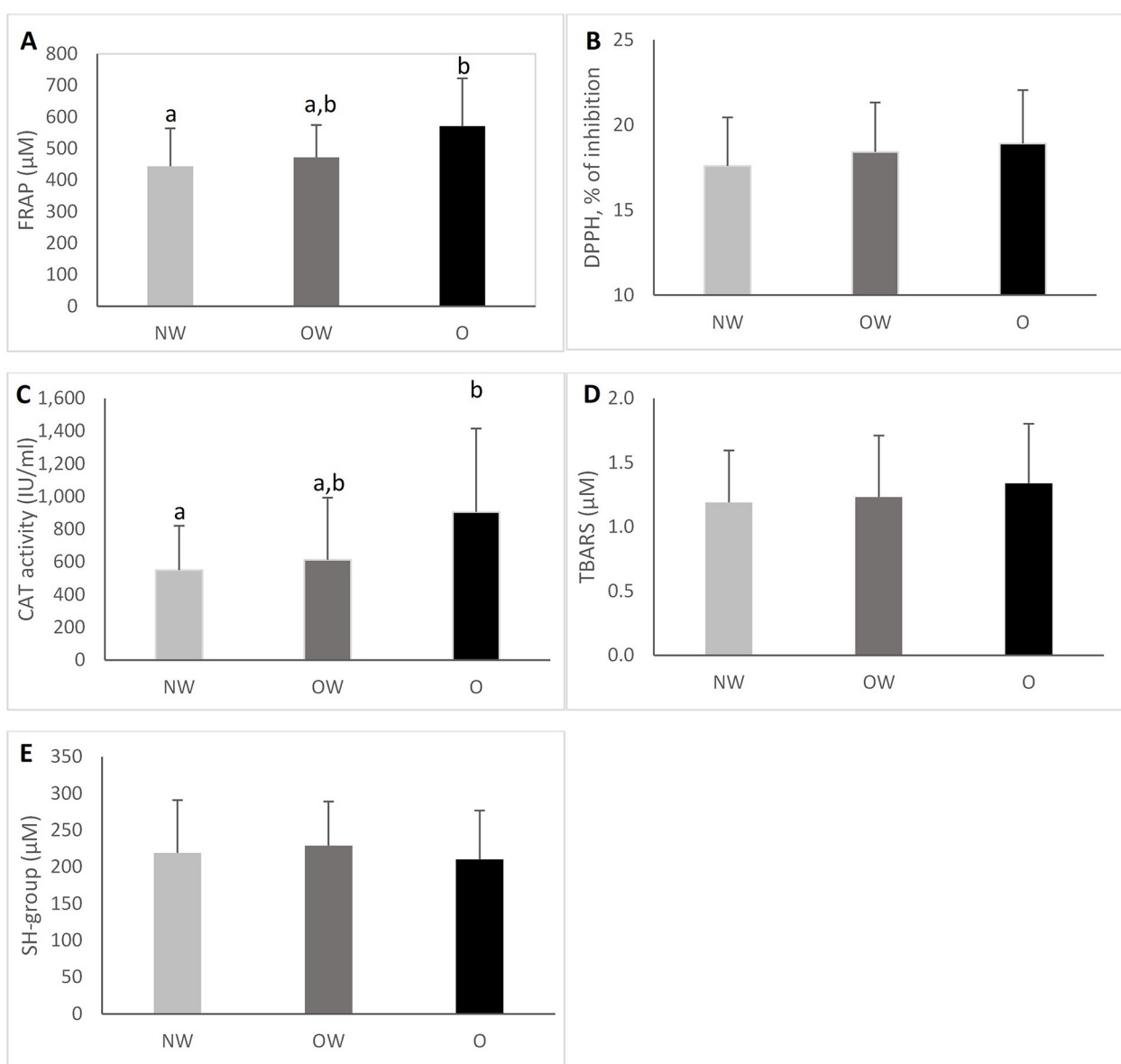

**Fig 1. Plasma antioxidant capacity and oxidant damage according to nutritional status.** (A) Ferric reducing ability of plasma (FRAP) activity. (B) 2,2-diphenyl-1-picrylhidrazyl radical (DPPH) (C) Catalase (CAT) activity. (D) Lipid peroxidation analysis, measured as malondialdehyde (MDA) level. (E) Protein thiol groups (SH-group); data are presented as mean and the error bar represents standard error. Different letters indicate significant differences between groups, where the O group has a higher antioxidant capacity, as measured by FRAP and CAT activity, compared to the NW group. The significance level was set at $^{*}p < 0.05$.

(MI), and model II (MII), adjusted by different confounders. Positive correlations were identified between uric acid and BMI (CM & MI p<0.001, MII p = 0.002), visceral fat (CM p <0.001, MI p = 0.002, MII p = 0.011), body fat (MI p<0.001, MII p = 0.001), total cholesterol (CM & MI p<0.001, MII p = 0.002), and triglycerides (CM & MI & MII p<0.001). Negative correlations were established between uric acid and HDL-cholesterol (CM & MI p<0.001, MII p = 0.004). Furthermore, uric acid was positively correlated with LDL-cholesterol in the crude model, (CM p = 0.013); however, when adjusted by confounders, the correlation was not maintained.

**Table 3. Correlations between oxidant/antioxidants parameters, body composition, and biochemical profile.**

| | | Visceral Fat (%) | Body Fat (%) | BMI (Kg/m$^2$) | Glucose (mg/dL) | Hba1c (%) | TC (mg/dL) | HDL (mg/dL) | LDL (mg/dL) | TG (mg/dL) | Creatinine (mg/dL) |
|---|---|---|---|---|---|---|---|---|---|---|---|
| Uric acid (mg/dL) | R | 0.507** | 0.106 | 0.447** | -0.115 | 0.047 | 0.447** | -0.408** | 0.251* | 0.564** | 0.473** |
| | p | <0.001 | 0.303 | <0.001 | 0.259 | 0.650 | <0.001 | 0.000 | 0.013 | <0.001 | <0.001 |
| FRAP (μM) | R | 0.404** | -0.028 | 0.352** | 0.074 | 0.183 | 0.303** | -0.160 | 0.240* | 0.428** | 0.344** |
| | p | <0.001 | 0.789 | 0.001 | 0.482 | 0.079 | 0.003 | 0.126 | 0.020 | <0.001 | 0.001 |
| DPPH (%) | R | 0.169 | -0.016 | 0.095 | 0.072 | 0.049 | 0.292** | -0.091 | 0.273** | 0.287** | 0.030 |
| | p | 0.115 | 0.880 | 0.368 | 0.496 | 0.645 | 0.005 | 0.387 | 0.009 | 0.006 | 0.778 |
| CAT activity (IU/mL) | R | 0.297** | 0.079 | 0.291** | 0.005 | 0.119 | 0.329** | -0.140 | 0.101 | 0.229* | 0.064 |
| | p | 0.005 | 0.456 | 0.005 | 0.961 | 0.259 | 0.001 | 0.184 | 0.336 | 0.028 | 0.543 |
| TBARS (μM) | R | 0.256* | 0.149 | 0.187 | 0.112 | -0.087 | 0.243* | -0.225* | 0.080 | 0.173 | -0.065 |
| | p | 0.018 | 0.167 | 0.080 | 0.298 | 0.415 | 0.022 | 0.034 | 0.455 | 0.104 | 0.542 |
| SH group (μM) | R | 0.019 | 0.039 | 0.002 | -0.122 | -0.034 | -0.161 | 0.188 | -0.195 | -0.217* | 0.043 |
| | p | 0.862 | 0.713 | 0.984 | 0.247 | 0.746 | 0.126 | 0.073 | 0.062 | 0.038 | 0.681 |

Natural logarithms were used for non-normal variables. Pearson's correlation test was applied. The significance level was established at

*$p < 0.05$

** $p < 0.01$. BMI: Body mass index; TC: Total cholesterol; HDL: HDL-cholesterol; LDL: LDL-cholesterol; TG: Triglycerides.

### Associations between antioxidant/oxidant parameters and plasma uric acid levels

Pearson correlation analyses were performed between uric acid levels and antioxidant/oxidant parameters across the study groups, as shown in **Fig 2**. We found significant correlations between uric acid levels and FRAP (R: 0.541, p < 0.001), and uric acid and CAT activity (R: 0.313, p = 0.002). After adjusting for confounders, the positive correlation remained as shown in **Table 4**. No other positive correlations were observed in the other groups and oxidant parameters.

In addition, linear regressions were applied to analyze the strength of the correlations, where FRAP (R: 0.514, β = 0.629, α = 4.929) and CAT activity (R: 0.313, β: 0.868, p = 0.002) were directly related to uric acid levels. We also independently analyzed the association in the OW + O group, where FRAP (R: 0.459, β = 0.578, α = 5.051) and CAT activity (R: 0.432, β = 0.868, α = 4.60) maintained the correlation with uric acid levels.

## Discussion

To the best of our knowledge, this is one of the few studies that first evaluates the antioxidant-oxidant parameters in the plasma of overweight and obese young adults and then analyzes the relationship between these parameters and plasma uric acid levels.

Concerning uric acid, body composition, and lipid profile, positive correlations were found between uric acid and BMI, visceral fat, body fat, total cholesterol, and triglycerides, while HDL cholesterol levels were negatively correlated. These results are consistent with previous studies that analyzed associations between plasma uric acid levels, obesity, and its metabolic consequences [3,4,16,17]. Given the dual nature of uric acid as both a pro-oxidant and an anti-oxidant, interpreting these results is complex. The net effect of uric acid on energy metabolism may vary depending on factors such as cellular redox status, the presence of other antioxidants, and tissue-specific conditions. Thus, during purine metabolism, the conversion of hypoxanthine to xanthine-by-xanthine oxidase generates superoxide anions, thereby increasing oxidative stress. This oxidative stress can contribute to mitochondrial dysfunction, leading to the

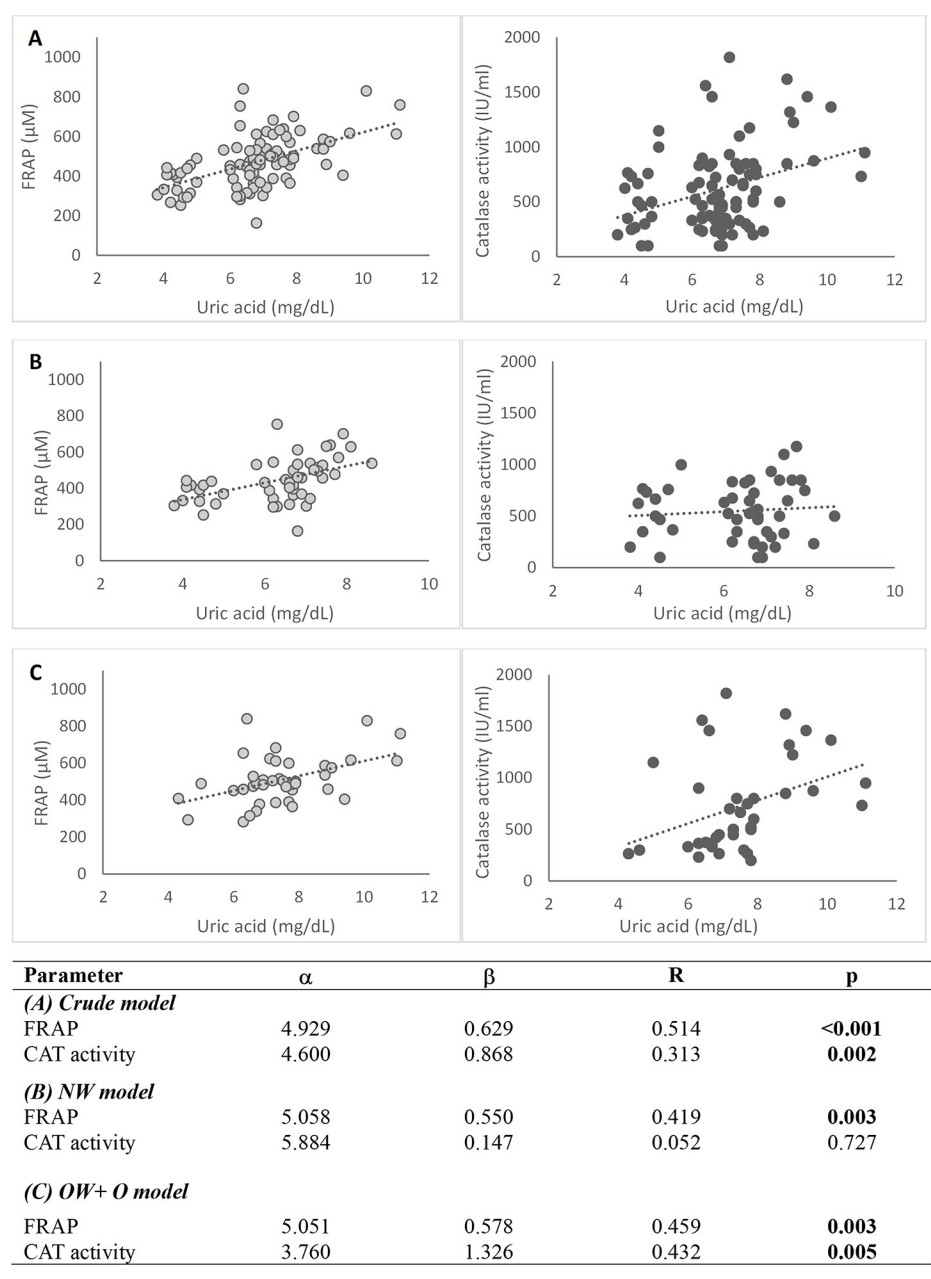

**Fig 2. Association between plasma uric acid levels and antioxidant status.** (A) Crude model: Linear regression between UA with FRAP (R: 0.514, p < 0.001) and CAT activity (R: 0.313, p = 0.002). (B) Model I: Linear regression between UA with FRAP (R: 0.419, p = 0.003) and CAT activity (R: 0.052, p = 0.727) in the NW group. (C) Model II: Linear regression between UA with FRAP (R: 0.459, p = 0.003) and CAT activity (R: 0.432, p = 0.005) in the OW + O group.

release of citrate into the cytoplasm, which subsequently promotes lipogenesis and triglyceride synthesis [18].

Furthermore, our results show significantly higher UA levels in the O group compared to the NW group (p = 0.001). Previous studies confirm this trend [3,4,9,13], and it is possibly due to the effect of obesity on uric acid metabolism. It can be hypothesized that excess body fat affects and increases nucleic acid metabolism, such as purines, stimulating uric acid

**Table 4. Correlations between uric acid and antioxidant/oxidant parameters.**

| Parameter | | Uric acid (mg/dL) | | |
|---|---|---|---|---|
| | | Crude Model | Model I | Model II |
| FRAP (μM) | R | 0.514** | 0.452* | 0.364* |
| | p | <0.001 | <0.001 | 0.002 |
| DPPH (%) | R | 0.175 | 0.127 | 0.103 |
| | p | 0.095 | 0.264 | 0.409 |
| CAT activity (IU/mL) | R | 0.313** | 0.358* | 0.351* |
| | p | 0.002 | 0.001 | 0.004 |
| TBARS (μM) | R | 0.185 | 0.184 | 0.170 |
| | p | 0.082 | 0.104 | 0.170 |
| SH group (μM) | R | 0.085 | 0.154 | 0.220 |
| | p | 0.419 | 0.176 | 0.074 |

** Indicates correlation is significant at $p < 0.01$.

* Indicates correlation is significant at $p < 0.05$.

Crude Model: No confounders were adjusted.

Model I: Sex, level of education, alcohol consumption, tobacco use, physical activity, and age were adjusted.

Model II: Sex, level of education, alcohol consumption, tobacco use, physical activity, age, and fruit, vegetable, white fish, and oily fish consumption were adjusted.

production. Moreover, obesity can cause damage in the glomerular hemodynamics, leading to an overactivation of the renin-angiotensin-aldosterone system and finally obesity-induced nephropathy. When these effects are long-lasting, glomerular atherosclerosis and insulin resistance can reduce the renal excretion of uric acid [22,23]. Additionally, there are determined adipocytokines, which play an important role in obesity and are correlated with the increase in uric acid levels. For instance, hyperuricemia is associated with low adiponectin and high leptin levels [24].

Conversely, when assessing the antioxidant potential of plasma in obese individuals, we observe elevated levels of CAT activity and FRAP scavenging compared to normal-weight individuals. This suggests that uric acid may exert antioxidant effects under certain conditions. Regarding CAT activity, our results agree with those presented by Adenan et. al., who likewise found an important increase in CAT activity in obese participants, which can be an indicator of oxidative stress [25]. Similarly, visceral fat, BMI, TC, and TG were positively correlated with CAT activity. Catalase is a major antioxidant enzyme that decomposes hydrogen peroxide into water and oxygen. In obesity, it has been proposed that the overexpression of catalase suppresses oxidative stress obesogenic pathways [26]. Specifically, it works against increased hydrogen peroxide production and proves to be beneficial in diminishing the effects of oxidative damage [25]. Furthermore, the linear regression analysis demonstrated that CAT activity was positively related to uric acid levels. Regarding this association, in a study conducted by Pallavi et al., uric acid activated the antioxidant system in stress-induced rat erythrocytes, increasing levels of catalase activity. They describe how uric acid plays an essential role in maintaining the enzymatic activity of catalase; therefore, an increase in uric acid levels could be translated into high catalase activity [27].

Respectively, results presented by Choromańska et al. and Chielle et al. confirm significantly higher values of FRAP activity in O groups compared to NW groups [1,13]. Our outcomes also showed positive correlations between FRAP and visceral fat, BMI, TC, LDL, TG, and creatinine, which are important biomarkers that are altered and increase in obese subjects (high BMI, visceral fat, and dyslipidemic profile) [1,13,28,29]. We hypothesize that the

mechanism that explains these correlations is related to uric acid, considered a vital player in the high antioxidant capacity of obese subjects. This hypothesis was supported by applying linear regressions, were FRAP activity was directly related to uric acid levels. This association remained after analyzing them independently in the OW + O group. Hence, Chielle et al. and Choromańska et al. confirm the positive correlation between FRAP and uric acid levels in obese participants [1,13]. The FRAP assay measures antioxidants with reducing power, such as uric acid and ascorbic acid. Particularly, FRAP measures the reduction of $Fe^{3+}$-TPTZ to $Fe^{2+}$-TPTZ. However, $Fe^{2+}$ is known to be a pro-oxidant compound that generates a superoxide radical from $H_2O_2$ through the Fenton reaction. For this reason, FRAP results assume that the antioxidants that can reduce $Fe^{3+}$-TPTZ also counteract the oxidative function of $Fe^{2+}$ [30]. In this regard, uric acid is one of the main compounds that have both abilities, thus explaining why FRAP assays may be highly dependent on plasmatic uric acid levels [10,30].

Moreover, this study measured uric acid levels in plasma, which representsextracellular uric acid. When this compound is found in this hydrophilic environment, it has antioxidant properties where it scavenges for ROS such as hydroxyl and peroxyl. This explains why obese individuals with higher uric acid levels in plasma display a higher antioxidant capacity, as measured by FRAP. In contrast, when uric acid is located within the cells, it exhibits well-known pro-oxidant actions. For example, it activates nicotinamide adenine dinucleotide phosphate oxidases (NADPH) involved in the generation of ROS, reduces endothelial levels of the antioxidant nitric oxide, activates peroxynitrite-mediated oxidation of lipids, stimulates proinflammatory biomarkers, and increases the activity of the xanthine oxidase enzyme, one of the major producers of ROS [6,31,32].

Even though DPPH analyzes total antioxidant capacity like FRAP, DPPH did not show a significant difference between the three groups. This is reasonable because, as Janaszewska et al. state, results that measure the same parameters through different mechanisms often vary notably. The main reason lies in the difference in reactivity of distinct antioxidants that contribute to the total antioxidant capacity of plasma with the corresponding indicators [33]. Furthermore, Lee et al. describe that the DPPH assay does not seem to be appropriate for determining plasma's total antioxidant capacity because the DPPH radical solution prepared with methanol may cause protein precipitation. To solve this, it is common to centrifuge the sample and use the supernatant to determine the absorbance. However, removing plasmatic proteins may hinder an accurate measurement of total antioxidant capacity since these proteins contribute from 10 to 50% to this capacity [30]. Nonetheless, DPPH was positively correlated with TC, LDL, and TG, agreeing with Jakubiak et al., because these markers reflect the metabolic disorders that characterize obesity [34].

Regarding the parameters that measure oxidative damage, although TBARS did not show significant differences between groups, there was a clear tendency for the damage to be greater in the obese group than the NW group. This matches the results presented by Adenan et al., which display the same trend, though theirs were not statistically meaningful [25]. Chielle et al., who did demonstrate a significant increase in TBARS in the obese group, explain that this effect is due to an increase in ROS that damages cell membranes. Likewise, TBARS kept a positive association with visceral fat and TC and a negative association with HDL cholesterol. Since obesity usually coincides with a dyslipidemic profile (high levels of LDL cholesterol and triglycerides, and low HDL cholesterol), the presence of ROS triggers lipid peroxidation [13].

Correspondingly, SH group levels seem to be lower in the O group compared to the NW group, though not significantly. In this regard, Chielle et al. were able to determine a significant reduction in plasma protein thiol groups when comparing obese participants to a normal weight control group. On the other hand, SH groups showed a negative correlation with TG, which coincides with Jakubiak et al. [34]. Plasma SH groups face oxidative molecules

responsible for protein damage, which are usually present in obese individuals. Smaller concentrations of SH in the obese group can be explained as a bigger consumption of thiol groups to counter an overproduction of ROS [13].

There are some limitations to our study that are worth mentioning. Firstly, the self-selection of participants and the single-center nature of the study limit our ability to determine the representativeness of our sample relative to the general Ecuadorian population; thus, selection bias could limit the generalizability of the results. Nevertheless, the sample is comparable to those from other studies [1,10–12,26], which allows us to compare our results. Another important aspect to consider is the handling of the biological samples, as these were stored for a considerable period before conducting the oxidant and antioxidant assays. However, we took measures to minimize any loss or alteration of the samples by storing them in aliquots at ultra-low temperatures and thawing them appropriately. Furthermore, we only considered some confounding variables when adjusting our analyses (sex, level of education, alcohol consumption, tobacco use, physical activity, age, and consumption of fruits, vegetables, white fish, and oily fish), which could create confounding bias. The study could not exclude participants on medications known to alter uric acid levels, such as diuretics. Consequently, we suggest that future studies should be conducted with a larger sample size and longitudinal design that can provide more definitive evidence. Those studies should consider other factors, such as medication or supplementation use, intake of all food groups, lipid concentration, and coexisting illnesses determined by biochemical analyses (such as diabetes mellitus) to achieve more accurate results. Finally, there are other markers that may be useful to complete the oxidant and antioxidant profile that we did not evaluate, such as superoxide dismutase (SOD), vitamin C, and tocopherol levels. Nonetheless, our findings follow a clear tendency that could only be confirmed through further analyses.

## Conclusions

The presented evidence suggests a positive association between parameters of antioxidant capacity (FRAP and CAT activity) and uric acid levels in overweight and obese individuals. Elevated plasma uric acid may exert a protective effect against obesity, implying a potential antioxidant function of uric acid in the pathogenesis of this condition. Although the underlying mechanisms still require further investigation, these associations could have relevant clinical implications for the understanding of oxidative stress and inflammation in obesity, as well as for the development of preventive and therapeutic strategies.

## Supporting information

**S1 Table. Consumption frequency of food groups by participants.** NW: Normal weight; OW: Overweight; O: Obese. The Chi-square test was applied to qualitative variables. The significance level was established at *$p < 0.05$. Data are presented as n (weighted %).
(DOCX)

**S2 Table. Correlations between plasma uric acid and lipid profile.** ** Indicates correlation is significant at $p < 0.01$. * Indicates correlation is significant at $p < 0.05$. Crude Model: No confounders were adjusted. Model I: Sex, level of education, alcohol consumption, tobacco use, physical activity, and age were adjusted. Model II: Sex, level of education, alcohol consumption, tobacco use, physical activity, age, and fruit, vegetable, white fish, and oily fish consumption were adjusted.
(DOCX)

## Acknowledgments

We extend our gratitude to the Escuela de Salud Pública (USFQ), especially to the Nutrition and Dietetics department for their pivotal role in facilitating the experimental component involving sample analysis. We extend our appreciation to Ms. Nataly Álava and Ms. Jessica Vivanco, laboratory technicians, for their significant contributions to the analytical procedures. Furthermore, we express our sincere gratitude to the participants who volunteered their time to participate in this study.

## Author Contributions

**Conceptualization:** Aida M. Chisaguano-Tonato, José M. Alvarez-Suarez.

**Data curation:** Aida M. Chisaguano-Tonato.

**Formal analysis:** Natalia Albuja-Quintana, Aida M. Chisaguano-Tonato, Maria E. Herrera-Fontana, Sandra Figueroa-Samaniego, José M. Alvarez-Suarez.

**Funding acquisition:** José M. Alvarez-Suarez.

**Investigation:** Natalia Albuja-Quintana, Aida M. Chisaguano-Tonato, Maria E. Herrera-Fontana, José M. Alvarez-Suarez.

**Methodology:** Aida M. Chisaguano-Tonato, Maria E. Herrera-Fontana, Sandra Figueroa-Samaniego, José M. Alvarez-Suarez.

**Project administration:** José M. Alvarez-Suarez.

**Resources:** Aida M. Chisaguano-Tonato, Maria E. Herrera-Fontana, Sandra Figueroa-Samaniego, José M. Alvarez-Suarez.

**Software:** Natalia Albuja-Quintana.

**Supervision:** Aida M. Chisaguano-Tonato, José M. Alvarez-Suarez.

**Validation:** Aida M. Chisaguano-Tonato, José M. Alvarez-Suarez.

**Visualization:** Natalia Albuja-Quintana, Aida M. Chisaguano-Tonato.

**Writing – original draft:** Natalia Albuja-Quintana, Sandra Figueroa-Samaniego.

**Writing – review & editing:** Aida M. Chisaguano-Tonato, Maria E. Herrera-Fontana, José M. Alvarez-Suarez.

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
