## [Decision Letter · Decision Letter 0]

6 Nov 2024

PONE-D-24-43131Relationship between plasma uric acid levels, antioxidant capacity, and oxidative damage markers in overweight and obese adults: A cross-sectional studyPLOS ONE

Dear Dr. Alvarez-Suarez,

Thank you for submitting your manuscript to PLOS ONE. After careful consideration, we feel that it has merit but does not fully meet PLOS ONE’s publication criteria as it currently stands. Therefore, we invite you to submit a revised version of the manuscript that addresses the points raised during the review process.

We look forward to receiving your revised manuscript.

Kind regards,

Aleksandra Klisic

Academic Editor

PLOS ONE

Journal Requirements:

“This work was supported by Universidad San Francisco de Quito (Grant Number: 15748) and by Grants from the School of Public Health of the Universidad San Francisco de Quito, Quito, Ecuador”

Reviewers' comments:

Reviewer's Responses to Questions

**Comments to the Author**

1. Is the manuscript technically sound, and do the data support the conclusions?

Reviewer #1: Partly

Reviewer #2: Yes

2. Has the statistical analysis been performed appropriately and rigorously? 

Reviewer #1: Yes

Reviewer #2: Yes

3. Have the authors made all data underlying the findings in their manuscript fully available?

Reviewer #1: Yes

Reviewer #2: Yes

4. Is the manuscript presented in an intelligible fashion and written in standard English?

Reviewer #1: Yes

Reviewer #2: Yes

5. Review Comments to the Author

Reviewer #1: In this study, the authors investigated the relationship between plasma uric acid levels, antioxidant capacity as well as oxidative damage markers in overweight and obese adults,which provided useful information. However, the manuscript is poorly presented (esp. the Discussion part) which hinders the readers to understand their work clearly. Here are some questions and suggestions about this manuscript.

1.They mentioned the volunteers were from the city of Machala, located at 6 m above sea level. What is the effects of “sea level” to the results?

2.“Participants were divided into three study groups according to Body Mass Index (BMI), specifically normal weight, pre-obese or overweight and obese.” What is the detailed definition of these three groups according to BMI and what is the justification of the definitions?

3.In the discussion part, the authors talk too much about the mechanisms which could not be supported by their own data such as Line 297 to 299 and Line 304 to 305. This is a cross-sectional study with limited numbers of volunteers. It can hardly explain the mechanisms without additional experimental data. It is better for the authors to reorganize the discussion part based on their own data to tell the readers what is new or what they really resolved in this study.

Reviewer #2: Overall, this is a well-constructed study with clear and scientifically sound objectives. The study’s novelty is highlighted by its focus on uric acid’s dual role in oxidative stress, specifically in young Ecuadorian adults. The authors demonstrate a solid understanding of biochemical analysis, and the statistical methods employed are rigorous and appropriate for the study’s objectives. Additionally, the participants are age-matched, and there appears to have been careful selection and assessment of subjects to ensure meaningful comparisons.

A few specific areas could be improved or clarified:

Lines 87-93: The language in this section would benefit from refinement to enhance scientific clarity. For example, using “centrifuged” instead of “centrifugated” would be more precise.

Line 135: It is unclear what "0.9 ± 0.02" refers to. This is likely an absorbance value, but clarification would help.

Line 138: The choice of methanol as a blank raises questions. Typically, a blank for DPPH should be the reaction mixture minus the test compounds to account for baseline absorbance without the antioxidants. Using plain DPPH as the blank may have impacted the accuracy of the results. Could the authors provide a rationale for this approach?

Figure 1: The use of indicators such as “a, a, b” on some graphs requires clarification—presumably, these are statistically significant differences, but a legend or explanation is necessary. Additionally, for biological measures like antioxidant activity or oxidative stress markers, presenting the data as Mean ± SD (instead of Mean ± SE) would provide a clearer picture of sample heterogeneity, particularly in biomedical contexts where individual variation is informative.

Technical Limitations: Although storage of samples at ultra-low temperatures minimizes degradation, prolonged storage could still affect the stability of certain biomarkers, particularly those sensitive to oxidative changes. It would be beneficial if the authors specified any precautions taken to prevent oxidative degradation during storage and handling.

Medication or Dietary Antioxidant Intake: It would be helpful to know if participants consumed any medication or dietary antioxidants, as these factors could influence oxidative stress markers and uric acid levels.

FRAP Assay: The study’s use of the FRAP assay is appropriate for comparing antioxidant levels across weight groups. However, additional controls would strengthen the claim that uric acid is the primary contributor to the observed changes in FRAP, rather than other antioxidant compounds.

Additionally, it would be valuable to explore potential lifestyle factors that might influence the results. For example, did the authors examine the effects of alcohol consumption or smoking on uric acid levels or antioxidant markers like catalase? Individuals who consume alcohol or smoke are often exposed to higher oxidative stress, which could impact both uric acid and catalase levels. Insight into whether catalase activity was lower among participants who smoked or drank compared to those who did not would enrich the analysis.

Lastly, while the study's cross-sectional design is appropriate for observing associations, a longitudinal design in future studies could provide a stronger basis for determining causal relationships between uric acid and oxidative stress markers over time.

6. PLOS authors have the option to publish the peer review history of their article (what does this mean?). If published, this will include your full peer review and any attached files.

Reviewer #1: No

Reviewer #2: No

---

## [Author Response · Author response to Decision Letter 0]

26 Nov 2024

Responses to Reviewers

Review Comments to the Author

Reviewer #1: In this study, the authors investigated the relationship between plasma uric acid levels, antioxidant capacity as well as oxidative damage markers in overweight and obese adults, which provided useful information. However, the manuscript is poorly presented (esp. the Discussion part) which hinders the readers to understand their work clearly. Here are some questions and suggestions about this manuscript.

1.They mentioned the volunteers were from the city of Machala, located at 6 m above sea level. What are the effects of “sea level” to the results?

The authors thank the reviewer for this observation. Regarding your question, we would like to point out that Ecuador has three distinct continental regions: the Coast, the Highlands (Sierra), and the Amazon, characterized by differences in altitude where their inhabitants live. For example, in Machala (the Coast), the population lives at sea level, whereas in the capital, Quito, we live at an altitude of 2,850 meters above sea level. Therefore, we consider it important to indicate, as a reference, the origin of the study sample since certain biochemical parameters (e.g., hemoglobin levels) and dietary factors (e.g., consumption of fish, fruits, vegetables, or tubers such as potatoes) are influenced by the altitude at which populations reside. These populations have well-defined habits based on the region in which they live. Thus, we deemed it necessary to clarify this aspect, as this particular characteristic of Andean countries is well-known and could potentially cause confusion.

In the interest of improving the description, we have updated the objective in the manuscript as follows:

….the purpose of this study was to evaluate the association between antioxidant capacity and oxidative damage markers with uric acid levels in overweight and obese young adults residing in Ecuador’s coastal region.

2.“Participants were divided into three study groups according to Body Mass Index (BMI), specifically normal weight, pre-obese or overweight and obese.” What is the detailed definition of these three groups according to BMI and what is the justification of the definitions?

As stated in the objectives of this research in the version currently under review, the purpose of our study was to evaluate the association between antioxidant capacity and oxidative damage markers with uric acid levels in overweight and obese adults. In this context, BMI remains a widely used indicator of body fat in epidemiological studies, facilitating the comparison of results across different populations and studies worldwide.

The classification presented here is based on the ranges established by the WHO, which defines three groups according to body weight as follows: normal weight (18.5 to 24.9 kg/m²), overweight (25.0 to 29.9 kg/m²), and obesity (≥30 kg/m²). To clarify this aspect and avoid confusion, we have included these ranges and the appropriate reference in the manuscript when describing each study group, along with the corresponding reference, as follows:

Participants were divided into three study groups according to Body Mass Index (BMI, kg/m2), specifically normal weight (NW, 18.5 to 24.9 Kg/m2; n=51), pre-obese or overweight (OW, 25 to 29.9 Kg/m2; n=27), and obese (O, � 30 Kg/m2; n=15).

Reference

Zierle-Ghosh A, Jan A. Physiology, Body Mass Index. In: StatPearls [Internet]. StatPearls Publishing; 5 Nov 2023 [cited 6 Nov 2024]. Available: https://www.ncbi.nlm.nih.gov/books/NBK535456/

3.In the discussion part, the authors talk too much about the mechanisms which could not be supported by their own data such as Line 297 to 299 and Line 304 to 305. This is a cross-sectional study with limited numbers of volunteers. It can hardly explain the mechanisms without additional experimental data. It is better for the authors to reorganize the discussion part based on their own data to tell the readers what is new or what they really resolved in this study.

The authors would like to thank the reviewer for their comments. In this regard, we would like to explain that the action mechanisms mentioned in the manuscript have served as a basis for providing a potential explanation for the results presented here. We agree with the reviewer that further studies with larger sample sizes are necessary to provide more specific insights into these mechanisms. However, we believe that the data presented here offer valuable information on the association between uric acid levels and the antioxidant capacity of plasma in this sample of the Ecuadorian population. This represents the first study of its kind conducted in the country.

While we acknowledge that the cross-sectional design and limited sample size restrict the ability to draw definitive causal conclusions, our findings suggest that uric acid may have a potential antioxidant role, in conjunction with catalase (CAT), as verified by the ferric reducing antioxidant power (FRAP) method. These results align with previous studies that have demonstrated higher uric acid levels and greater antioxidant capacities in obese individuals. We have discussed these findings throughout the manuscript, including the potential pathways through which uric acid might exert its effects. Thus, we have relied on these possible causal mechanisms to interpret the results presented here, which we believe is necessary for contextualizing our findings. Additionally, we are aware of the limitations of our study, and in the manuscript, we have emphasized the need for further research to elucidate the underlying mechanisms. Future studies with larger sample sizes and longitudinal designs could provide more definitive evidence.

In our opinion, our study contributes to the understanding of uric acid’s dual role in oxidative and antioxidant processes and underscores the need for further research to explore its complex interactions with conditions such as obesity. We also believe it is important to highlight that our results provide a valuable first insight into these phenomena in populations from Andean countries. Notably, future research is already underway to explore potential variations in these findings, taking into account the altitudinal factor of these populations.

Reviewer #2: Overall, this is a well-constructed study with clear and scientifically sound objectives. The study’s novelty is highlighted by its focus on uric acid’s dual role in oxidative stress, specifically in young Ecuadorian adults. The authors demonstrate a solid understanding of biochemical analysis, and the statistical methods employed are rigorous and appropriate for the study’s objectives. Additionally, the participants are age-matched, and there appears to have been careful selection and assessment of subjects to ensure meaningful comparisons.

A few specific areas could be improved or clarified:

Lines 87-93: The language in this section would benefit from refinement to enhance scientific clarity. For example, using “centrifuged” instead of “centrifugated” would be more precise.

The authors thank the reviewer for their observation and recommendation. Following their suggestions, we have revised the description of the participant selection process in our study to make it more technical. Please refer to lines 87–116 of the updated manuscript.

Ninety-three volunteers (28 men and 65 women, aged 22.24 ± 4.5 years) from the city of Machala in the province of El Oro, Ecuador, were included in this study. Between January 9th and February 14th, 2020, biological samples, sociodemographic data, lifestyle factors, and physical measurements were collected from each selected participant. The inclusion criteria used to accept participants were: adults aged 18 to 50 years living in the selected areas of Machala from birth until the time of the study, who agreed to participate and signed the informed consent. Volunteers could not be pregnant or suffer from chronic diseases (diabetes, cancer, or cardiovascular diseases) or immunological diseases (autoimmune or immunosuppressive). Additionally, subjects who had spent time in places with a difference in altitude greater than 2,000 meters were not included. 

Participants were divided into three study groups according to Body Mass Index (BMI, kg/m2), specifically normal weight (NW, 18.5 to 24.9 Kg/m2; n=51), pre-obese or overweight (OW, 25 to 29.9 Kg/m2; n=27), and obese (O, � 30 Kg/m2; n=15) [15]. The study design was cross-sectional, and it was conducted in accordance with the principles of the Declaration of Helsinki, as revised in 2000. The protocol was approved by the Human Research Ethics Committee (CEISH) of the Universidad San Francisco de Quito (Code P2018-176E), Ecuador, and registered at the General Coordination of Strategic Health Development of the Ministry of Public Health of Ecuador with protocol code MSPCURI000308-2. Written consent was obtained from each participant. To analyze and present the findings of this study, we accessed a dedicated database created specifically for this research between January 2023 and July 2024. To maintain participant privacy and confidentiality, each individual was assigned a unique identifier code throughout the data analysis process.

Line 135: It is unclear what "0.9 ± 0.02" refers to. This is likely an absorbance value, but clarification would help.

The authors thank the reviewer for this observation. The absorbance values mentioned here refer to the absorbance values of a DPPH solution at its maximum absorption. This value is subsequently used for calculating antioxidant activity, considering the degree of discoloration, and therefore the reduction in absorbance values, produced by the reaction between the DPPH solution and the sample (plasma).

To clarify this sentence and avoid confusion we have rewritten this part as follows:

DPPH solution was prepared by mixing 3.2 mg of DPPH radical with 100 mL of absolute methanol, and this solution was adjusted to a maximum absorbance of 0.9�0.02 to 517 nm. This absorbance was used to calculate the antioxidant activity according to the following formula, in accordance with the method used. 

DPPH radical scavenging activity was calculated as:

% DPPH radical scavenging activity = [(CA - SA) / CA] × 100

Where:

CA = Absorbance of DPPH solution with methanol

SA = Absorbance of DPPH solution in the sample solution (plasma).

Line 138: The choice of methanol as a blank raise question. Typically, a blank for DPPH should be the reaction mixture minus the test compounds to account for baseline absorbance without the antioxidants. Using plain DPPH as the blank may have impacted the accuracy of the results. Could the authors provide a rationale for this approach?

We understand the reviewer's concern regarding this matter. However, we would first like to point out that, for this analysis, we used a protocol previously reported for this type of plasma analysis according to Prymont-Przyminska et al. (2014). Additionally, in our opinion, the use of methanol as a blank is justified, as the calculation of the sample's antioxidant activity requires a reference DPPH solution. This reference solution is then used to calculate the antioxidant activity by measuring the degree of discoloration caused by the reaction between the sample and the DPPH solution. This DPPH solution corresponds to the one previously mentioned in response to the reviewer's question about the absorbance of "0.9 ± 0.02". If a blank were prepared using the DPPH solution without the sample, it would not be possible to use it as a reference to determine the percentage of discoloration of the DPPH solution caused by the sample, as was explained in the previous answer. 

Prymont-Przyminska A, Zwolinska A, Sarniak A, Wlodarczyk A, Krol M, Nowak M, et al. Consumption of strawberries on a daily basis increases the non-urate 2,2-diphenyl-1-picryl-hydrazyl (DPPH) radical scavenging activity of fasting plasma in healthy subjects. J Clin Biochem Nutr. 2014;55: 48–55. doi:10.3164/JCBN.13-93

Figure 1: The use of indicators such as “a, a, b” on some graphs requires clarification—presumably, these are statistically significant differences, but a legend or explanation is necessary. Additionally, for biological measures like antioxidant activity or oxidative stress markers, presenting the data as Mean ± SD (instead of Mean ± SE) would provide a clearer picture of sample heterogeneity, particularly in biomedical contexts where individual variation is informative.

Thank you very much for the suggestion. We have included a more detailed explanation of the use of letters to indicate the differences between the study groups in the figure caption. Although Figure 1 originally used the standard error as a measure of dispersion which, as we know, is also a valid metric for representing variability, we have updated the figure to include the standard deviation for each group. This change better reflects the interindividual variability within our sample. Following your suggestions, we have clarified this aspect in the text as follows:

The total antioxidant capacity, as measured by FRAP scavenging and CAT activity, was higher in the obese group compared to the normal weight group (p= 0.007 and p= 0.038, respectively), as shown in Fig. 1. However, no significant differences were observed in these parameters between the OW and O groups. Also, there were no significant differences in levels of plasma DPPH, TBARS, and SH group between the groups.

Moreover, we have detailed in the figure legend all the information regarding the comparisons between groups as follows:

Fig. 1. Plasma antioxidant capacity and oxidant damage according to nutritional status (A) Ferric reducing ability of plasma (FRAP) activity. (B) 2,2-diphenyl-1-picrylhidrazyl radical (DPPH). (C) Catalase (CAT) activity. (D) Lipid peroxidation analysis, measured as malondialdehyde (MDA) level. (E) Protein thiol groups (SH group); data are presented as mean and the error bar represents standard error. Different letters indicate significant differences between groups, where the O group has a higher antioxidant capacity, as measured by FRAP and CAT activity, compared to the NW group. The significance level was set at *p < 0.05.

Technical Limitations: Although storage of samples at ultra-low temperatures minimizes degradation, prolonged storage could still affect the stability of certain biomarkers, particularly those sensitive to oxidative changes. It would be beneficial if the authors specified any precautions taken to prevent oxidative degradation during storage and handling.

We understand the reviewer’s concern in this regard and agree with their observation. This was an aspect which we paid close attention to. The samples were stored under identical conditions at -80°C. As the reviewer noted, ultra-low temperatures minimize degradation. To prevent potential sample degradation caused by repeated thawing and refreezing during analysis, the samples were divided into 0.5 mL plasma aliquots. For each analysis, the required number of aliquots was fully used, and any remaining samples were discarded. Additionally, the thawing process was always performed at 4°C and in the dark. This procedure was applied consistently for all samples and across all types of analyses. By doing so, we ensured that all samples were subjected to the same conditions and procedures, minimizing the risk of degradation and maintaining uniform handling.

To address this point, we have included a more detailed description of the handling of plasma aliquots in the manuscript as follows (Please refer to lines 140–143):

To maintain sample integrity and prevent degradation from repeated freeze-thaw cycles, plasma samples were divided into 0.5 mL aliquots. For each analytical run, the necessary number of aliquots was thawed at 4°C in the dark to minimize temperature fluctuations and light exposure.

Medication or Dietary Antioxidant Intake: It would be helpful to know if participants consumed any medication or dietary antioxidants, as these factors could influence oxidative stress markers and uric acid levels.

The authors agree with the reviewer on this point. The use of medications and possible supplementation in this age group was qualitatively assessed thr

---

## [Decision Letter · Decision Letter 1]

9 Dec 2024

Relationship between plasma uric acid levels, antioxidant capacity, and oxidative damage markers in overweight and obese adults: A cross-sectional study

PONE-D-24-43131R1

Dear Dr. Alvarez-Suarez,

We’re pleased to inform you that your manuscript has been judged scientifically suitable for publication and will be formally accepted for publication once it meets all outstanding technical requirements.

Kind regards,

Aleksandra Klisic

Academic Editor

PLOS ONE

Additional Editor Comments (optional):

Reviewers' comments:

Reviewer's Responses to Questions

**Comments to the Author**

1. If the authors have adequately addressed your comments raised in a previous round of review and you feel that this manuscript is now acceptable for publication, you may indicate that here to bypass the “Comments to the Author” section, enter your conflict of interest statement in the “Confidential to Editor” section, and submit your "Accept" recommendation.

Reviewer #1: All comments have been addressed

Reviewer #2: All comments have been addressed

2. Is the manuscript technically sound, and do the data support the conclusions?

Reviewer #1: (No Response)

Reviewer #2: (No Response)

3. Has the statistical analysis been performed appropriately and rigorously? 

Reviewer #1: (No Response)

Reviewer #2: (No Response)

4. Have the authors made all data underlying the findings in their manuscript fully available?

Reviewer #1: (No Response)

Reviewer #2: (No Response)

5. Is the manuscript presented in an intelligible fashion and written in standard English?

Reviewer #1: (No Response)

Reviewer #2: (No Response)

6. Review Comments to the Author

Reviewer #1: (No Response)

Reviewer #2: (No Response)

7. PLOS authors have the option to publish the peer review history of their article (what does this mean?). If published, this will include your full peer review and any attached files.

Reviewer #1: No

Reviewer #2: No

---

## [Editor Report · Acceptance letter]

8 Jan 2025

PONE-D-24-43131R1 

PLOS ONE

Dear Dr. Alvarez-Suarez, 

I'm pleased to inform you that your manuscript has been deemed suitable for publication in PLOS ONE. Congratulations! Your manuscript is now being handed over to our production team.

Kind regards, 

on behalf of

Dr. Aleksandra Klisic 

Academic Editor

PLOS ONE